# Comparative Transcriptomic Analyses for the Optimization of Thawing Regimes during Conventional Cryopreservation of Mature and Immature Human Testicular Tissue

**DOI:** 10.3390/ijms25010214

**Published:** 2023-12-22

**Authors:** Cheng Pei, Plamen Todorov, Mengyang Cao, Qingduo Kong, Evgenia Isachenko, Gohar Rahimi, Nina Mallmann-Gottschalk, Pamela Uribe, Raul Sanchez, Volodimir Isachenko

**Affiliations:** 1Department of Obstetrics and Gynecology, Medical Faculty, Cologne University, 50931 Cologne, Germany; cheng.pei@uk-koeln.de (C.P.); qingduokong@gmail.com (Q.K.); evgenia.isachenko@uk-koeln.de (E.I.); nina.mallmann-gottschalk@uk-koeln.de (N.M.-G.); 2Institute of Biology and Immunology of Reproduction of Bulgarian Academy of Sciences (BAS), 1113 Sofia, Bulgaria; plamen.ivf@gmail.com; 3Medizinisches Versorgungszentrum AMEDES für IVF- und Pränatalmedizin in Köln GmbH, 50968 Cologne, Germany; 4Center of Excellence in Translational Medicine, Scientific and Technological Bioresource Nucleus (CEMT-BIOREN), Temuco 4810296, Chile; pamela.uribe@ufrontera.cl (P.U.); raul.sanchez@ufrontera.cl (R.S.); 5Department of Internal Medicine, Faculty of Medicine, Universidad de la Frontera, Temuco 4811230, Chile; 6Department of Preclinical Sciences, Faculty of Medicine, Universidad de la Frontera, Temuco 4811230, Chile

**Keywords:** human, testicular tissue, cryopreservation, thawing, RNA sequencing, transcriptomics, differentially expressed genes (DEG), Kyoto Encyclopedia of Genes and Genomes (KEGG), gene ontology (GO), protein–protein interactions (PPI)

## Abstract

Cryopreservation of human testicular tissue, as a key element of anticancer therapy, includes the following stages: saturation with cryoprotectants, freezing, thawing, and removal of cryoprotectants. According to the point of view existing in “classical” cryobiology, the thawing mode is the most important consideration in the entire process of cryopreservation of any type of cells, including cells of testicular tissue. The existing postulate in cryobiology states that any frozen types of cells must be thawed as quickly as possible. The technologically maximum possible thawing temperature is 100 °C, which is used in our technology for the cryopreservation of testicular tissue. However, there are other points of view on the rate of cell thawing, according to how thawing should be carried out at physiological temperatures. In fact, there are morphological and functional differences between immature (from prepubertal patients) and mature testicular tissue. Accordingly, the question of the influence of thawing temperature on both types of tissues is relevant. The purpose of this study is to explore the transcriptomic differences of cryopreserved mature and immature testicular tissue subjected to different thawing methods by RNA sequencing. Collected and frozen testicular tissue samples were divided into four groups: quickly (in boiling water at 100 °C) thawed cryopreserved mature testicular tissue (group 1), slowly (by a physiological temperature of 37 °C) thawed mature testicular tissue (group 2), quickly thawed immature testicular tissue (group 3), and slowly thawed immature testicular tissue (group 4). Transcriptomic differences were assessed using differentially expressed genes (DEG), the Kyoto Encyclopedia of Genes and Genomes (KEGG), gene ontology (GO), and protein–protein interaction (PPI) analyses. No fundamental differences in the quality of cells of mature and immature testicular tissue after cryopreservation were found. Generally, thawing of mature and immature testicular tissue was more effective at 100 °C. The greatest difference in the intensity of gene expression was observed in ribosomes of cells thawed at 100 °C in comparison with cells thawed at 37 °C. In conclusion, an elevated speed of thawing is beneficial for frozen testicular tissue.

## 1. Introduction

Fertility protection of humans and endangered species has always been a hot topic, and spermatozoa cryopreservation has always been a standard method for the preservation of male and animal fertility [1]. However, this technique is not suitable for adolescents and prepubertal individuals who have not yet fully developed gonads. Because they cannot produce mature spermatozoa, testicular tissue cryopreservation is a better fertility preservation technique. For children diagnosed with cancer, the 5-year survival rate has increased to 80%; however, 25% of male survivors still have azoospermia, and testicular tissue cryopreservation is very important for these patients [2,3,4]. Some adult males require gonadotoxic treatment due to cancer, autoimmune diseases (e.g., systemic lupus erythematosus and systemic sclerosis), and genetic disorders (sickle cell disease thalassemia and idiopathic medulla aplasia). Cryopreservation of testicular tissue for these patients is also an alternative to spermatozoa cryopreservation [5].

Conventional cryopreservation (slow or programmable freezing) is the most commonly used method for testicular tissue preservation [6]. Spermatogenesis is not only related to testicular germ cells but also to testicular somatic cells (e.g., Sertoli cells and Leydig cells) and even the extracellular matrix. The testicular transcriptome represents the sum of all transcripts expressed by different cell groups in testicular tissue. Therefore, RNA sequencing provides the ability to study the entire process of spermatogenesis and the development of testicular cells. Previously published results, including sequencing data, showed that the testis is the organ with the most tissue-specific genes [7,8]. However, there are relatively few RNA sequencing reports on cryopreserved testicular tissue, and only reports on cryopreserved testicular tissue of mice and cats [9,10] are published.

Cryopreservation of human testicular tissue includes the following stages: saturation of cells with permeable cryoprotectants, freezing, thawing, and removal of these cryoprotectants from the cells. As per the perspective prevalent in “classical” cryobiology, the thawing mode holds crucial importance in the cryopreservation technique for all cell types, including testicular tissue cells.

The existing postulate in cryobiology states that any frozen types of cells must be thawed as quickly as possible. The technologically maximal possible thawing rate can be realized with the thawing of cells in boiling water (at 100 °C). This mode of thawing was used in our experiments in comparison with the thawing at physiological temperature (37 °C).

However, there are other points of view on the rate of cell thawing, according to which thawing should be carried out at physiological temperatures. In fact, there are morphological and functional differences between immature (from prepubertal patients) and mature testicular tissue. Accordingly, the question of the influence of thawing temperature on both types of tissues is relevant.

The purpose of this study is to explore the transcriptomic differences of cryopreserved mature and immature testicular tissue subjected to different thawing methods by RNA sequencing.

## 2. Results

Following the appropriate thawing and removal of cryoprotectants (Figure 1), the cell viability was assessed.

### 2.1. Differentially Expressed Genes (DEG)

At the beginning of the described research, volcano maps were drawn to compare up- and down-regulated differentially expressed genes in tissues of different groups. In comparison with cells of group 2 (slowly thawed mature), 219 differentially expressed genes (DEG) in cells of group 1 (quickly thawed mature) were up-regulated, and 437 DEGs were down-regulated (Figure 2A). At the same time, in comparison with group 2 (slowly thawed mature), the expression of 1073 DEGs in cells of group 4 (slowly thawed immature) was increased, and the expression of 2942 DEGs was decreased (Figure 2B). In comparison with cells of group 1 (quickly thawed mature), 4976 DEGs in group 3 (quickly thawed immature) were up-regulated, and 6071 DEGs were down-regulated. Group 3 has the most intensive DEG expression (Figure 2C). In cells of groups 3 (quickly thawed immature) and 4 (slowly thawed immature), cells which had no significant changes, only one up-regulated and down-regulated DEG was found (Figure 2D).

In contrast with cells of “mature” groups 1 (quickly thawed) and 2 (slowly thawed), cells of “immature” groups 3 (quickly thawed) and 4 (slowly thawed) had 3919 up-regulated DEGs and 6461 down-regulated DEGs (Figure 2E).

### 2.2. Differentially Expressed Genes (DEG) through Enrichment Analysis of Kyoto Encyclopedia of Genes and Genomes (KEGG) Pathways

In comparison with cells of group 2 (slowly thawed mature tissue), cells of group 1 (quickly thawed mature) are mainly enriched in ribosomes, coronavirus disease COVID-19, rheumatoid arthritis, leishmaniasis, and phagosomes (Figure 3A). Compared to group 2, DEGs in group 4 (slowly thawed immature cells) are mainly enriched in ribosomes, coronavirus disease COVID-19, oocyte meiosis, progesterone-mediated oocyte maturation, and glycolysis/gluconeogenesis (Figure 3B). DEGs in cells of groups 1 and 3 (quickly thawed immature) are mainly enriched in ribosomes, coronavirus disease COVID-19, oocyte meiosis, cellular senescence, and progesterone-mediated oocyte maturation (Figure 3C).

KEGG enrichment analysis of cells from these groups was similar, with ribosomes and COVID-19-related pathways ranked in the top two. DEGs in groups 3 and 4 were mainly enriched in pathways related to oxidative stress, including the NF-κB signaling pathway, the fox O signaling pathway, the p53 signaling pathway, and apoptosis (Figure 3D). Finally, cells from “immature” groups 3 and 4 and from “mature” groups 1 and 2 showed different KEGG pathways, ribosomes, cell cycle, coronavirus disease–COVID-19, glycosaminoglycan degradation, and oocyte meiosis (Figure 3E).

### 2.3. Differentially Expressed Genes (DEG) through Gene Ontology (GO) Enrichment Analysis

Compared to cells from group 2 (slowly thawed mature), cells of group 1 (quickly thawed mature) are mainly enriched in cytoplasmic translation, SRP-dependent cotranslational protein targeting to membrane, viral transcription, nuclear-transcribed mRNA catabolic process, nonsense-mediated decay, and translational initiation (Figure 4A). The main differences between the GO pathway in cells of group 2 (slowly thawed mature) and group 4 (slowly thawed immature) are spermatogenesis, SRP-dependent co-translational protein targeting to membrane, cytoplasmic translation, nuclear-transcribed mRNA catabolic process, nonsense-mediated decay, and viral transcription (Figure 4B). Compared to cells from group 1 (quickly thawed mature), the GO pathway in cells of group 3 (quickly thawed immature) is mainly reflected in spermatogenesis, cytoplasmic translation, SRP-dependent co-translational protein targeting to membrane, cell cycle, and viral transcription (Figure 4C).

GO analysis of cells from these three groups is relatively similar. The GO pathway in cells of group 3 (quickly thawed immature) and group 4 (slowly thawed immature) demonstrated obsolete activation of MAPKKK activity, chromatin silencing, heterochromatin formation, positive regulation of p38MAPK cascade, and negative regulation of protein kinase activity (Figure 4D).

GO pathways between cells from “immature” groups 3 (quickly thawed immature) + 4 (slowly thawed immature) and cells from “mature” groups 1 (quickly thawed mature) + 2 (slowly thawed mature) include spermatogenesis, cytoplasmic translation, SRP-dependent co-translational protein targeting to membrane, viral transcription, nuclear-transcribed mRNA catabolic process, and nonsense-mediated decay (Figure 4E). The biological process of spermatogenesis is the most enriched, indicating that spermatogenesis in testicular tissue at different developmental stages is more important than the stress caused by different ways of thawing (Figure 3B,C,E). The cellular component and molecular function enrichment of the GO pathway are presented in the Appendix A.

### 2.4. Protein–Protein Interactions (PPI) Network

In cells of groups 1 (quickly thawed mature) and 2 (slowly thawed mature), the ribosome-related protein family has more link nodes; the most important is RPS27A, with a total of 28 link sites, and the remaining RPS14, RPS3A, RPS4X, and RPS6 are also important. PRM2, SPATA3, TNP1, SPATA16, and TNP2 are the more prominent proteins in comparison between cells of groups 2 (slowly thawed mature) and 4 (slowly thawed immature). In the PPI network of cells from groups 1 (quickly thawed mature) and 3 (quickly thawed immature), CSF1R with 22 link sites is the most prominent, and TYROBP, RPS27A, STAT1, and RPS14 are also important (Figure 5).

## 3. Discussion

At present, relevant sequencing studies have been conducted on human cryopreserved spermatozoa, proving that cryopreservation of spermatozoa is a genetically safe fertility preservation method. Moreover, cryopreservation by direct plunging into liquid nitrogen (vitrification) may reduce negative biological changes in spermatozoa in comparison with traditional controlled (conventional) freezing [11]. In our research, RNA sequencing was used to analyze the transcriptional differences produced by different thawing methods in cryopreserved mature and immature testicular tissue in order to optimize the cryopreservation protocol.

### 3.1. Differentially Expressed Genes (DEG)

It was established that DEG expression in cells from group 3 (quickly thawed immature) was less compared to cells from group 4 (slowly thawed immature). The up-regulated genes were H2AC19, and the down-regulated were GADD45B. This indicates that the immature testicular tissue had less of a response to both different protocols of thawing (slow and quick). H2AC19 is a member of the histone H2A family; its related biochemical processes include the regulatory mechanisms of RNA polymerase I promoter opening and telomere end packaging [12,13]. Recent studies have shown that ubiquitination and acetylation of histone H2A may lead to spermatogenesis disorders in males [14]. The same difference in H2AC19 expression was also observed in cells of groups 2 (slowly thawed mature) and 4 (slowly thawed immature). GADD45B belongs to the growth arrest and DNA damage-induced 45 (GADD45) gene family, whose family members also include GADD45A and GADD45G. These genes are related to physiological and environmental stress and mainly regulate cell proliferation, apoptosis, and DNA damage [15,16]. GADD45B plays a biological role by binding and activating MTK1/MEKK4 kinase and then affecting the p38/JNK pathway [17]. It was reported that stress of epithelial cells due to storage at different temperatures causes changes in GADD45B expression [18].

For quick and slow thawing, the expression of DEGs in cells from groups 1 (quickly thawed mature) and 3 (quickly thawed immature) was much more intensive than that in cells from groups 2 (slowly thawed mature) and 4 (slowly thawed immature), with the number of DEGs being 11,047. It shows that by the quick thawing, the transcriptomic changes of immature cells are more expressed than those of mature cells.

### 3.2. Kyoto Encyclopedia of Genes and Genomes (KEGG) Pathways

The KEGG pathway enrichment in cells of groups 3 (quickly thawed immature) and 4 (slowly thawed immature) with few DEGs showed that the NF-κB signaling pathway was highly enriched. This pathway is mainly involved in inflammatory and immune responses, and its family members are NF-κB1, NF-κB2, Rel A, Rel B, and c-Rel [19]. During the stress of testicular tissue, NF-κB in Sertoli cells is activated, leading to germ cell apoptosis [20]. Ranked second in KEGG pathway enrichment is the Fox O signaling pathway, which is also mainly involved in apoptosis and oxidative stress. During oxidative stress, such as increased reactive oxygen species, activation of c-Jun *N*-terminal kinase (JNK) is induced, which leads to translocation of Fox O phosphorylation in the cytoplasm, ultimately affecting downstream targets [21,22]. KEGG analysis showed that cryopreserved immature testicular tissue is intolerant to quick thawing compared to slow thawing.

Except for cells from groups 3 (quickly thawed immature) and 4 (slowly thawed immature), which have fewer DEGs, KEGG pathway enrichment analysis of cells showed that the changes in the ribosome pathway were the most observed. Ribosomes are the site of protein synthesis within the cell, translating specific genetic information into proteins via messenger RNA. The ribosome-associated pathway includes ribosomal components as well as small nucleolar RNA (snoRNA), ribosomal proteins, and non-ribosomal proteins. The ribosome pathway can cause reproductive toxicity by regulating gene expression, leading to spermatogenesis disorders [23,24]. In addition to occurring during protein synthesis, ribosome collisions often occur when cells are under stress. When cells are subjected to different stresses, it will also cause ribosomes to collide with each other [25]. Current research shows that when cells are subjected to moderate stress, ribosome collision causes the activation of the GCN2-mediated integrated stress response (ISR) signaling pathway. When cells are subjected to high-intensity stress, high-intensity ribosome collisions will activate the p38/JNK-guided signaling pathway [26,27]. However, it is not yet clear which signaling pathway will be activated by cellular stress caused by the different thawing methods used in our research.

The second pathway with significant changes is coronavirus disease (COVID-19). Because some tissue samples were collected during the COVID-19 epidemic, it is uncertain whether the patient was already infected when the sample was collected or whether there was a possibility of contamination during cryopreservation. COVID-19 causes lung damage in patients by binding to the high-affinity angiotensin-converting enzyme 2 (ACE-2) receptors [28]. However, ACE-2 is also abundantly expressed in testicular tissue, especially Leydig cells and spermatogonia. The most abundant protein in Leydig cells is insulin-like factor 3 (INSL3), and its expression was also found to be most significantly decreased in the testicular tissue of infected patients [29]. Pathological examination of testicular tissue of patients who died of serious illness found that the seminiferous epithelium became thinner and the number of apoptotic cells in the seminiferous tubules increased, indicating that infection with COVID-19 can cause spermatogenesis disorders in patients [30].

### 3.3. Gene Ontology (GO)

Overall, the GO analysis showed that the highest enrichment of spermatogenesis was due to different developmental stages of testicular tissue. The remaining most abundant factor is cytoplasmic translation, which is a reaction in which ribosomes mediate protein formation in the cytoplasm. To ensure the stability of the protein state, control mechanisms of the cell’s co-translational quality are completed by regulating related mRNA, recycling ribosomes, and degrading nascent polypeptide chains [31]. SRP-dependent co-translational protein targeting the membrane is also a highly enriched biological process. The newly synthesized polypeptide chain usually carries an *N*-terminal hydrophobic signal sequence. When the polypeptide chain appears in the ribosome polypeptide exit channel, it will be recognized and bound by SRP and then transported to the endoplasmic reticulum. This pathway can minimize nascent primary proteins misfold and aggregate before reaching the endoplasmic reticulum [32,33].

In relation to our research, different thawing methods will mainly affect biological processes by affecting various stages of protein translation.

### 3.4. Protein–Protein Interactions (PPI)

PPI mapping by DEG cells of groups 1 (quickly thawed mature) and 2 (slowly thawed mature) showed that RPS27A was the most important protein. RPS27A is mainly located in the middle and tail of the spermatozoon and regulates spermatozoon motility. The expression of RPS27A is downregulated in patients with asthenozoospermia and patients exposed to oxidative stress (reactive oxygen species), indicating that it may be a protein marker for detecting spermatozoon motility [34,35]. The difference between cells of groups 1 (quickly thawed mature) and 2 (slowly thawed mature) is mainly due to the application of two different thawing methods. This indicates that RPS27A, which also has great changes caused by temperature stress. The name of PRM2 is Protamine-2, and Protamine-1 can often replace it.

Therefore, maintaining a stable ratio of PRM2 and PRM1 is important to protect the stability of spermatozoon DNA and to reduce the impact of oxidative stress [36,37]. In our study, it was shown that PRM2 plays an important role in the cryopreservation of testicular tissue. CSF1R, which has 22 linking sites, is more active in testicular tissue of different ages by quick thawing. These data are similar to data from previous research regarding increasing CSF1R in mouse testicular tissue with age [38].

A comprehensive analysis using KEGG, GO, and PPI showed that oxidative stress-related pathways and ribosome-mediated protein translation-related pathways play an important role in testicular tissue at different stages and with the use of different thawing methods. Due to the limitation of testicular tissue collection, it cannot be ruled out that the specificity of individual testicular tissue may affect the analysis of sequencing results. In fact, the study of testicular tissue taking into account the development of spermatozoa with the culture of cells in vitro for a certain period with the following sequencing is more informative.

### 3.5. Some Practical Aspects of Described Technology for Cryopreservation

The most common cryo-injuries during thawing include cell osmotic shock and recrystallization. Ice crystals initially melt in the extracellular fluid, causing a decrease in osmotic pressure relative to the intracellular environment. This leads to an influx of extracellular water, resulting in cellular swelling and eventual disintegration. Simultaneously, water molecules entering the cells can also promote the growth of intracellular ice crystals and cell damage, especially if the thawing rate is slow [39].

The survival of cryopreserved cells is contingent on the warming rate during thawing, and cell death associated with intracellular ice formation is primarily attributed to ice recrystallization rather than the initial nucleation of ice [40,41].

Utilizing a rapid thawing method at 100 °C in boiling water allows us to apply the principles of the following cryobiological concept. According to this concept, any biological specimen preserved using any existing cryopreservation techniques should be thawed as swiftly as possible.

In our laboratory conditions, this involves immersing the specimen in boiling water. In theory, an alternative warming medium like boiling oil (ranging from 250 to 300 °C) could be used, resulting in a thawing rate several times faster than in boiling water. However, the use of boiling oil for thawing is unsuitable in the sterile environment of a reproductive laboratory.

An essential aspect of thawing in boiling water is the agitation of the water. When a cryo-vial containing a frozen specimen with a temperature of −150 to −130 °C is exposed to room temperature for 30 s and then placed in unstirred water, a layer of water with a temperature significantly lower than 100 °C forms between the cryo-vial wall and the boiling water. This layer acts as an insulator, reducing the thawing speed. By using a magnetic stirrer to agitate the boiling water, the cool layers of water on the cryo-vial surface are continuously replaced by water with a temperature of +100 °C. Our calculations estimate that this element of technology increases the speed of thawing by 15–20%. Additionally, exposing the cryo-vial with biomaterial extracted from liquid nitrogen, which is inherently non-sterile, serves as a surface sterilization step for the cryo-vial, which is crucial in medical technology.

Traditional cryopreservation protocols, which aim to protect cells and prevent intracellular crystallization during sub-zero cooling, typically involve the use of permeable cryoprotectants. These cryoprotectants often include three high molecular alcohols such as glycerol, ethylene glycol, propylene glycol, and dimethyl sulfoxide (DMSO). These components constitute 10 to 12% of the total solution, typically comprising either DMSO or a combination of DMSO and one of the glycols [42].

In our protocol, we used multi-cryoprotectants, similar to how we protect ovarian tissue cells. Our data suggest that the protective effect of 12% DMSO alone was inferior to that of a 12% solution supplemented with a combination of cryoprotectants (data not published).

## 4. Materials and Methods

### 4.1. Design of Experiments

A total of 12 human testicular tissue samples (Figure 6) were collected and divided into 4 groups: quickly (in boiling water at 100 °C) thawed cryopreserved mature testicular tissue (group 1), slowly (by physiological temperature 37 °C) thawed mature testicular tissue (group 2), quickly thawed immature testicular tissue (group 3), and slowly thawed immature testicular tissue (group 4). Mature testicular tissues were collected from 6 adults, and immature testicular tissues were collected from 2 children. In each experimental group, 3 samples were used.

### 4.2. Extraction and Cryopreservation of Testicular Tissue (Equilibration with Cryoprotectants, Thawing and Removal of Cryoprotectants)

The study was conducted in accordance with the Declaration of Helsinki and approved by the Institutional Ethics Committee of Cologne University (protocols 01-106, 12–163, 17-427, 20-1229, code BioMSOTE) and the Bulgarian National Medical Institutional Ethics Committee (Project “Development of new cryopreservation methods to restore testicular function in adult and prepubertal patients with oncological diseases”, approval No. 7-021/2022). The informed consent was obtained from patients whose testicular tissue was collected for this study. All chemicals were obtained from Sigma (Sigma Chemical Co., St. Louis, MO, USA) unless otherwise stated.

All patients underwent a testicular biopsy after the diagnosis of azoospermia as well as for fertility preservation before initiating any therapy carrying a high risk of permanent infertility, such as high-dose chemotherapy. Testicular tissue was obtained from 2 boys and 6 adults aged 3 and 5 and from 34 to 41, respectively.

The procedure of extraction of testicular tissue has been previously described in detail [43,44,45,46,47,48,49,50,51]. Briefly, a midline incision was made in the scrotum and the testis, and the spermatic cord was removed, preferably from the hemiscrotum, with the larger testis. Tunica vaginalis was opened, and Tunica albuginea was visualized. Under an operating microscope, Tunica albuginea was widely opened in an equatorial plane, preserving the subtunical vessels. After the opening of Tunica albuginea, testicular parenchyma was examined directly at 12-fold magnification under the operating microscope. Small samples (1–18 mg) were excised by pulling out larger, more opaque tubules from surrounding Leydig cell nodules or hyperplasia in the testicular parenchyma.

Cryopreservation of testicular tissue (Figure 1) was performed according to the previously published protocol for human ovarian tissue [52,53,54,55]. Collected testicular fragments were equilibrated for 30 min. in a cryopreservation solution containing 6% dimethyl sulfoxide, 6% ethylene glycol, and 0.15 M sucrose. The cryovials were then placed in an Ice Cube 14S freezer (SyLab, Neupurkersdorf, Austria) for conventional freezing. The freezing procedure is as follows: When the cryovial reaches −7 °C, start the freezing process and cool at a rate of −0.3 °C per minute until it reaches −33 °C. The process usually takes about 90 min. Finally, the cryo-vials were placed in liquid nitrogen for long-term storage. Tissue was thawed with different regimes.

*Quick thawing:* Thawing of tissue was achieved by holding the cryo-vial for 30 s at room temperature, followed by immersion in a 100 °C (boiling) water bath for 60 s, and expelling the contents of the vial into the solution for the removal of cryoprotectants. The exposure time in the boiling water was visually controlled by the presence of ice in the medium; as soon as the ice reached size 2 to 1 mm, the vial was removed from the boiling water, at which point the final temperature of the medium was between 4 and 10 °C. Within 5 to 10 s after thawing, the tissue fragments from the cryo-vials were expelled into a 10 mL thawing solution (basal medium containing 0.5 M sucrose) in a 100 mL specimen container (Sarstedt, Nuembrecht, Germany). After the exposure of the tissues to sucrose for 15 min, stepping rehydration of cells was performed, as reported previously [52,53,54,55].

*Slow thawing:* This thawing regime was exactly the same as the quick thawing regime described above, except that the tissues were thawed by immersing the cryovial in a 37 °C water bath for three minutes.

### 4.3. Sequencing and Data Extraction

Each sample of testicular tissue was used for RNA extraction with the Trizol method. It was detected that the RIN/RQN of all samples was greater than 4.

Strand-specific transcriptome library construction was completed by enriching mRNA from total RNA, sequenced by DNBSEQ high-throughput platform, and followed by bioinformatics analysis. The library was validated on the Agilent Technologies 2100 bioanalyzer. The library was amplified with phi29 to make a DNA nanoball (DNB), which had more than 300 copies of one molecule. The DNBs were loaded into the patterned nanoarray, and single-end 50 (pair-end 100/150) base reads were generated using combinatorial Probe-Anchor Synthesis (cPAS). RNA-seq analysis was performed using the Dr. Tom System (https://biosys.bgi.com, accessed on 25 August 2023). The raw data of RNA-seq number is BioProject: PRJNA1030294. It can be downloaded at “Sequence read archive” on the National Center for Biotechnology Information (https://www.ncbi.nlm.nih.gov/bioproject/1030294, accessed on 14 March 2024). Because the raw data for sequencing contains reads of low quality, adapter contamination, and excessively high levels of unknown base N, these reads need to be removed before data analysis to ensure the reliability of the results.

### 4.4. Differentially Expressed Genes (DEG) Analysis

DEG is the abbreviation of genes, which refers to the detection of genes with different expression levels in different samples. Map clean reads were run to a reference gene sequence (transcriptome), and then gene expression levels for each sample were calculated. Detection of DEG was performed by the DEseq2 method. This DEseq2 method is based on the principle of negative binomial distribution. Our project uses the previously described method [56]. It was analyzed first and maps all candidate genes to each entry in the Gene Ontology database (http://www.geneontology.org/, accessed on 25 August 2023).

R’s basis function phyper was used (https://stat.ethz.ch/R-manual/R-devel/library/stats/html/Hypergeometric.html, accessed on 25 August 2023) to calculate the *p*-value. Then the *p*-value is corrected through multiple tests, and the corrected package is the q-value (https://bioconductor.org/packages/release/bioc/html/qvalue.html, accessed on 25 August 2023). Finally, q-value (corrected *p*-value) ≤ 0.05 was used as the threshold, and the KEGG and GO term that satisfied this condition was defined as the KEGG and GO term that was significantly enriched in candidate genes. PPI analysis of differentially expressed genes was based on the STRING database with known and predicted protein-protein interactions.

## 5. Conclusions

No fundamental differences in the quality of cells of mature and immature testicular tissue after cryopreservation were found. Generally, thawing of mature and immature testicular tissue was more effective at 100 °C. The greatest difference in the intensity of gene expression was observed in ribosomes of cells thawed at 100 °C in comparison with cells thawed at 37 °C. In conclusion, an elevated speed of thawing is beneficial for frozen testicular tissue.

## Figures and Tables

**Figure 1 ijms-25-00214-f001:**
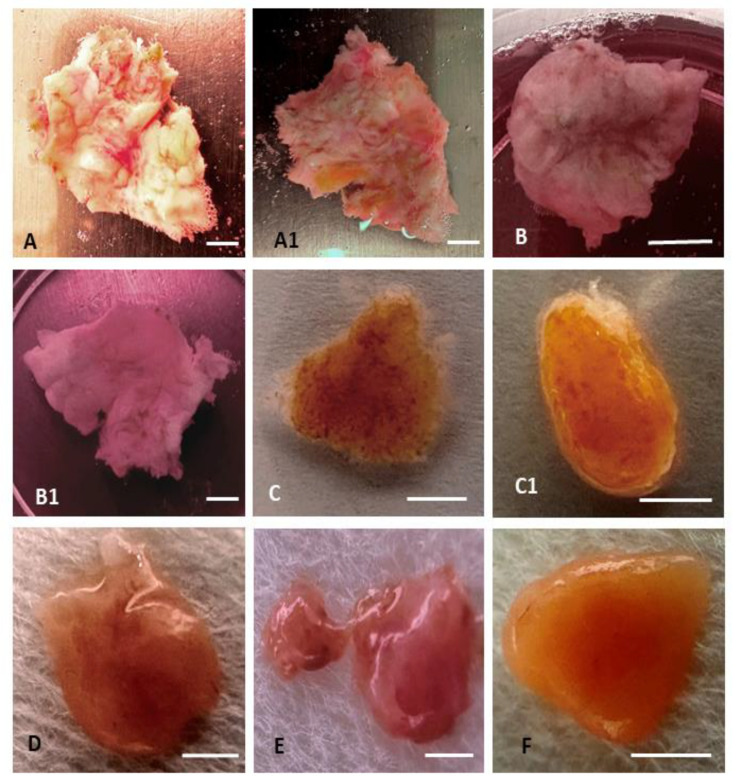
Fragments of mature and immature testicular tissue after cryopreservation. (**A**) Fragment of mature testicular tissue of patient B. from the *tunica albuginea side* (“outer” layer) after cryopreservation. (**A1**) The same fragment of tissue of patient B. from the *seminiferous tubules* side (“inner” layer). (**B**) Fragment of mature testicular tissue from patient D. in a hypertonic solution (0.5 M sucrose) during removal of cryoprotectants shows tissue compaction as a result of cell dehydration. (**B1**) The same fragment of testicular tissue of patient D. in isotonic solution after removal of cryoprotectants. (**C**) Fragment of immature testicular tissue from patient C. after cryopreservation. (**C1**) Demonstration photographs: the same dehydrated fragment on filter paper of immature testicular tissue from patient C. after cryopreservation. (**D**–**F**) Demonstration photographs: dehydrated fragments on filter paper of mature testicular tissue from patients K., P., and M. after cryopreservation. Scale Bar = 1 mm.

**Figure 2 ijms-25-00214-f002:**
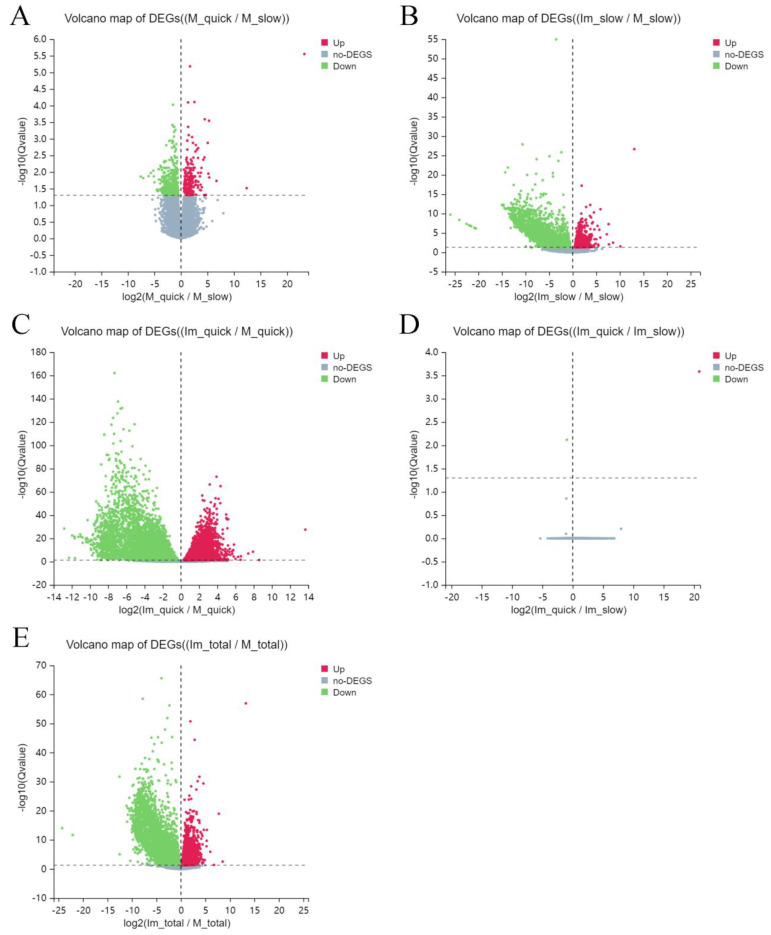
Volcano map of differentially expressed genes (DEG) after different regimes of thawing. (**A**) DEG volcano map: cells of group 1 (quickly thawed mature) vs. cells of group 2 (slowly thawed mature). (**B**) DEG volcano map: cells of group 4 (slowly thawed immature) vs. cells of group 2 (slowly thawed mature). (**C**) DEG volcano map: cells of group 3 (quickly thawed immature) vs. cells of group 1 (quickly thawed mature). (**D**) DEG volcano map: cells of group 3 (quickly thawed immature) vs. cells of group 4 (slowly thawed immature). (**E**) DEG volcano map: cells of group 3 (quickly thawed immature) + group 4 (slowly thawed immature) vs. cells of group 1 (quickly thawed mature) + group 2 (slowly thawed mature).

**Figure 3 ijms-25-00214-f003:**
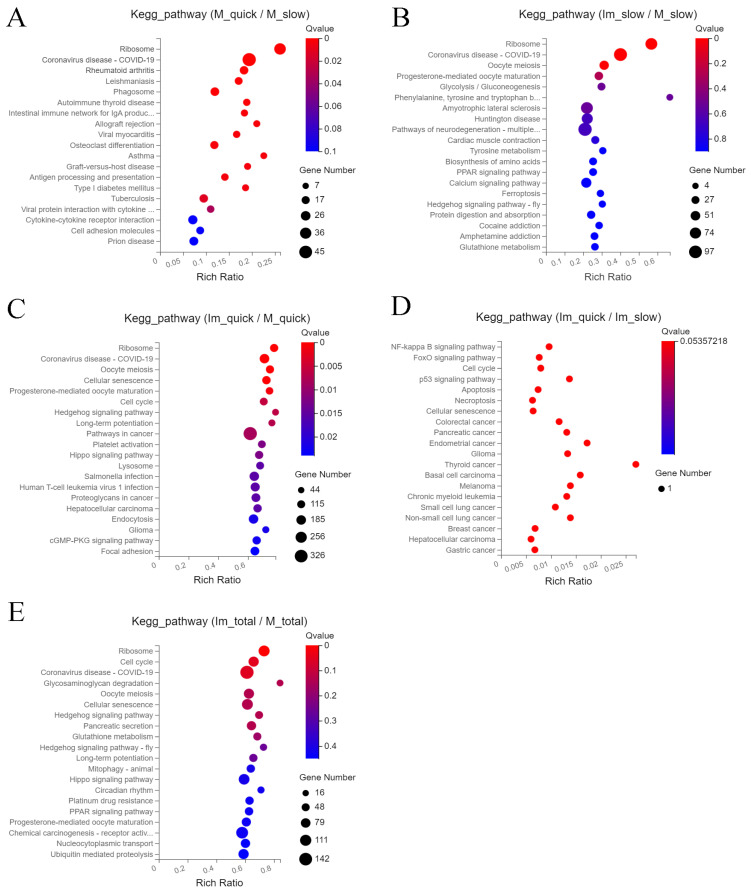
Kyoto Encyclopedia of Genes and Genomes (KEGG) pathway enrichment bubble chart of Differentially Expressed Genes (DEG). (**A**) KEGG pathway chart of DEG in cells of groups 1 (quickly thawed mature) and 2 (slowly thawed mature). (**B**) KEGG pathway chart of DEGs in cells of groups 2 (slowly thawed mature) and 4 (slowly thawed immature). (**C**) KEGG pathway chart of DEGs in cells of groups 1 (quickly thawed mature) and 3 (quickly thawed immature). (**D**) KEGG pathway chart of DEGs in cells of groups 3 (quickly thawed immature) and 4 (slowly thawed immature). (**E**) KEGG pathway chart of DEGs in cells of groups 3 (quickly thawed immature) + 4 (slowly thawed immature) and groups 1 (quickly thawed mature) + 2 (slowly thawed mature).

**Figure 4 ijms-25-00214-f004:**
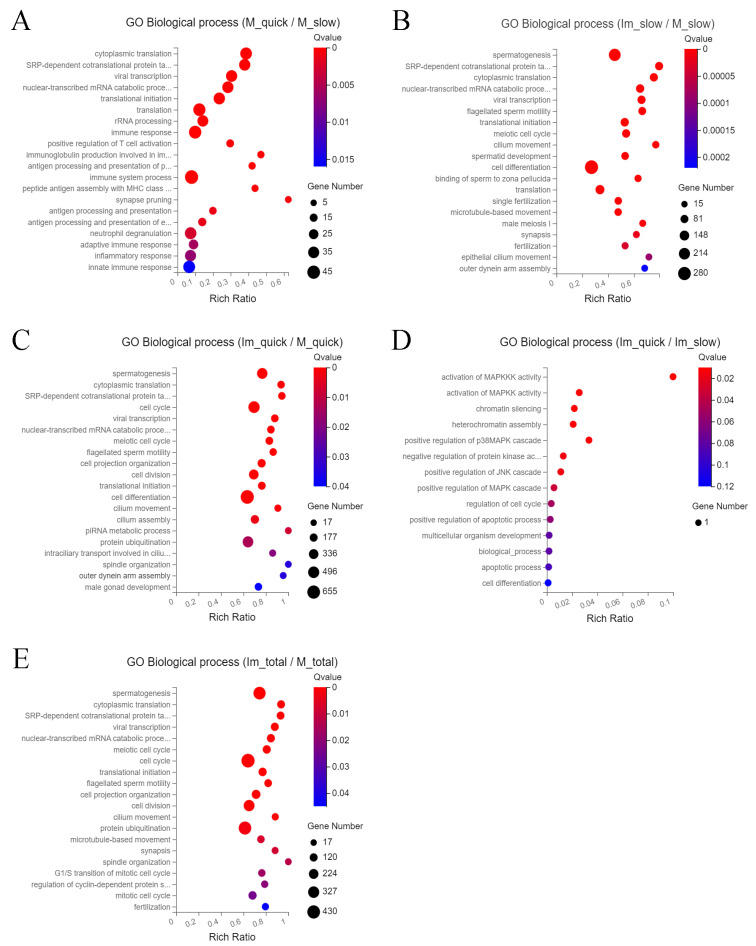
Visualization of Gene Ontology (GO) enrichment bubble chart for cells of different groups. (**A**) GO enrichment bubble chart for groups 1 (quickly thawed mature) and 2 (slowly thawed mature). (**B**) GO enrichment bubble chart for groups 2 (slowly thawed mature) and 4 (slowly thawed immature). (**C**) GO enrichment bubble chart for groups 1 (quickly thawed mature) and 3 (quickly thawed immature). (**D**) GO enrichment bubble chart for groups 3 (quickly thawed immature) and 4 (slowly thawed immature). (**E**) GO enrichment bubble chart for groups 3 (quickly thawed immature) + 4 (slowly thawed immature) and groups 1 (quickly thawed mature) + 2 (slowly thawed mature).

**Figure 5 ijms-25-00214-f005:**
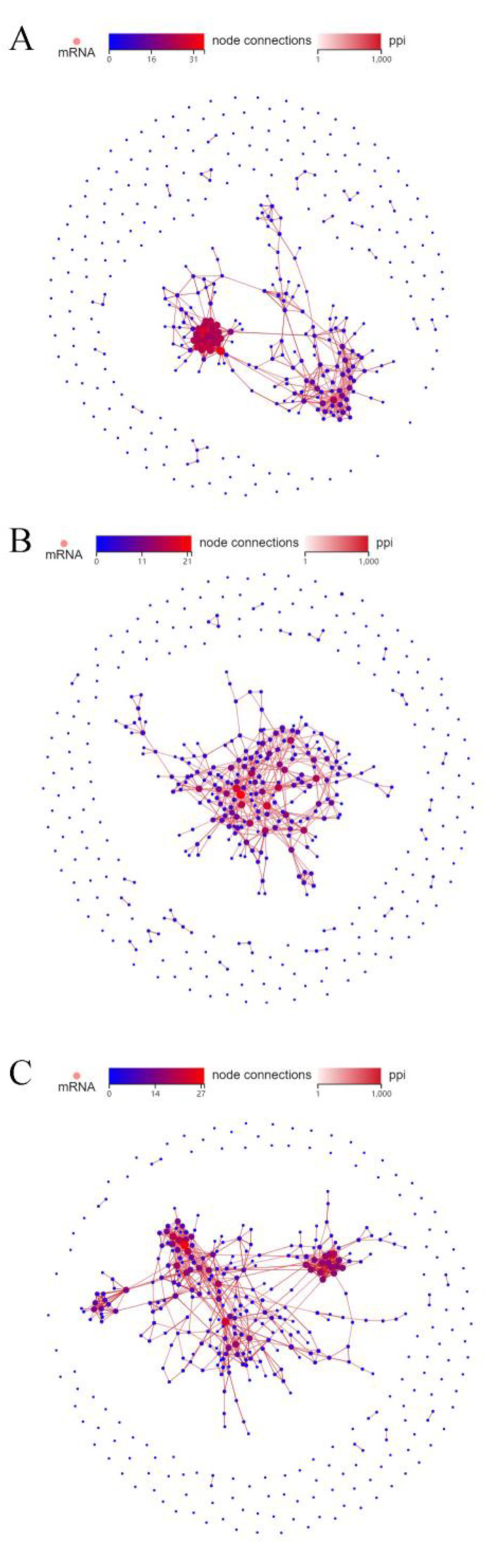
Protein–protein interactions (PPI) networks. (**A**) PPI network for cells of groups 1 (quickly thawed mature) and 2 (slowly thawed mature). (**B**) PPI network for cells of groups 2 (slowly thawed mature) and 4 (slowly thawed immature). (**C**) PPI network for cells of groups 1 (quickly thawed mature) and 3 (quickly thawed immature).

**Figure 6 ijms-25-00214-f006:**
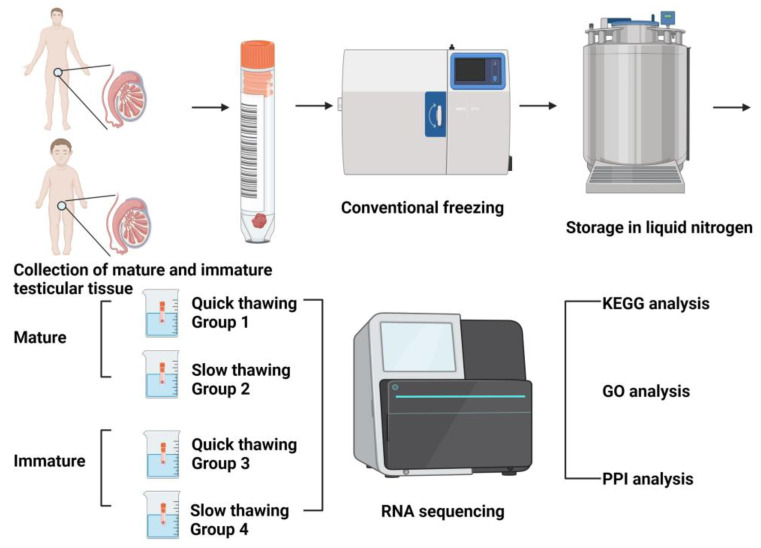
Design of experiments. (KEGG) Kyoto Encyclopedia of Genes and Genomes, (GO) Gene Ontology, (PPI) protein–protein interaction.

## Data Availability

The raw data of RNA-seq can be downloaded at “Sequence read archive” on National Center for Biotechnology Information (https://www.ncbi.nlm.nih.gov/bioproject/1030294, accessed on 14 March 2024).

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
