# Peer review of "Comparative Transcriptomic Analyses for the Optimization of Thawing Regimes during Conventional Cryopreservation of Mature and Immature Human Testicular Tissue"

_ijms, 2023, doi:10.3390/ijms25010214_

Round 1
Reviewer 1 Report
Comments and Suggestions for Authors
Strategies to assure the highest quality of reproductive structures following cryopreservation is currently a pressing topic in andrology. Most of currently available papers focus on spermatozoa, nevertheless, testicular tissue cryopreservation is equally as important, particularly in certain cohorts of patients. In this sense, the paper deals with an important theme, and could be of interest for specialists in reproductive medicine and cryobiology. In principle, the study is well designed, and the data are novel. Nevertheless, I would recommend considering several editions to the paper in order to increase its potential.
In the Introduction section, I would recommend adding some thoughts on transcriptomics as scientific direction with great promise in andrology and what observations have been recorded so far using this approach in andrology (testicular tissue or spermatozoa), and then interconnect it with the experimental objectives to the study.
The Material and methods need further clarification: How was the testicular tissue obtained? Out of the subjects, how may were adults and how many were infants? What was the age range of the patients? How was RNA extracted out of the tissue?
The Discussion section could benefit from a summary of the most important differences observed, translated into a take-home message for the reader. Also, limitations should be properly discussed.
The authors mention in the Abstract and Conclusion that statistically significant differences were observed amongst thawing at 37°C and 100°C, however no statistical analysis is detailed in the actual main body of the manuscript, and I see no graph or figure depicting the values for such conclusion. Please, explain further.
Finally, please revise the list of references according to the instruction for authors.
Author Response
Answer to Reviewer 1.
Reviewer 1: Strategies to assure the highest quality of reproductive structures following cryopreservation is currently a pressing topic in andrology. Most of currently available papers focus on spermatozoa, nevertheless, testicular tissue cryopreservation is equally as important, particularly in certain cohorts of patients. In this sense, the paper deals with an important theme, and could be of interest for specialists in reproductive medicine and cryobiology. In principle, the study is well designed, and the data are novel. Nevertheless, I would recommend considering several editions to the paper in order to increase its potential.
Authors: Dear Sir/madam,
Thank you very much for your careful work with our manuscript.
All your suggestions and advices are accepted and respective corrections are made in manuscript.
Reviewer 1: In the Introduction section, I would recommend adding some thoughts on transcriptomics as scientific direction with great promise in andrology and what observations have been recorded so far using this approach in andrology (testicular tissue or spermatozoa), and then interconnect it with the experimental objectives to the study.
Authors: Thank you very much for your suggestion. Now we have written in Introduction:
Conventional cryopreservation (slow freezing, programmable freezing) is the most common using method for testicular tissue preservation [6]. Spermatogenesis is not only related to testicular germ cells, but also to testicular somatic cells (e.g. Sertoli cells, Leydig cells), and even the extracellular matrix. The testicular transcriptome represents the sum of all transcripts expressed by different cell groups in testicular tissue. Therefore, RNA sequencing allows better to study the entire process of spermatogenesis and development of testicular cells. Previously published results, included sequencing data, showed that the testis is the organ with the most tissue-specific genes [7, 8]. However, there are relatively few RNA sequencing reports on cryopreserved testicular tissue, and only reports on cryopreserved testicular tissue of mice and cats [9, 10] are published.
Reviewer 1: The Material and methods need further clarification: How was the testicular tissue obtained? Out of the subjects, how may were adults and how many were infants? What was the age range of the patients? How was RNA extracted out of the tissue?
The Discussion section could benefit from a summary of the most important differences observed, translated into a take-home message for the reader. Also, limitations should be properly discussed.
Authors: Thank you very much. Thank you for your right suggestion. Now in our manuscript we have written (in Introduction, Materials and Methods as well as in Discussion):
The testicular transcriptome represents the sum of all transcripts expressed by different cell groups in testicular tissue. Therefore, RNA sequencing allows better to study the entire process of spermatogenesis and development of testicular cells. Previously published results, included sequencing data, showed that the testis is the organ with the most tissue-specific genes [7, 8]. However, there are relatively few RNA sequencing reports on cryopreserved testicular tissue, and only reports on cryopreserved testicular tissue of mice and cats [9, 10] are published.
The technologically maximal possible thawing rate can be realized with the thawing of cells in boiling water (at 100°C). Just this mode of thawing was used in our experiments in comparison with the thawing at physiological temperature (37°C).
Mature testicular tissues were collected from 6 adults, immature testicular tissues were collected from 2 children. In each experimental group it was used 3 samples.
All patients were undergoing testicular biopsy after diagnosis of azoospermia as well as for fertility preservation before initiating any therapy carrying a high risk of permanent infertility, such as high-dose chemotherapy. Testicular tissue was obtained from 2 boys and 6 adults aged 3 and 5 and from 34 to 41, respectively.
The procedure of extraction of testicular tissue has been previously described in detail [43-51]. Briefly, a midline incision was made in scrotum and testis with spermatic cord was removed preferably from the hemiscrotum with the larger testis. Tunica vaginalis was opened and tunica albuginea was visualized. Under an operating microscope, tunica albuginea was widely opened in an equatorial plane preserving the subtunical vessels. After opening of tunica albuginea, testicular parenchyma was examined directly at 12-fold magnification under the operating microscope. Small samples (1–18 mg) were excised by pulling out larger, more opaque tubules from surrounding Leydig cell nodules or hyperplasia in the testicular parenchyma.
Cryopreservation of testicular tissue (Figure 1) was performed according to the previously published protocol for human ovarian tissue [52-61].
Each sample of testicular tissue was used for RNA extraction with the Trizol method. It was detected that the RIN/RQN of all samples was greater than 4.
Strand specific transcriptome library construction was completed by enriching mRNA from total RNA, sequenced by DNBSEQ high-throughput platform, and followed by bioinformatics analysis. Library was validated on the Agilent Technologies 2100 bioanalyzer. The library was amplified with phi29 to make DNA nanoball (DNB) which had more than 300 copies of one molecular. The DNBs were load into the patterned nanoarray and single end 50 (pair end 100/150) bases reads were generated in the way of combinatorial Probe-Anchor Synthesis (cPAS). RNA-seq analysis was performed using the Dr. Tom System (https://biosys.bgi.com). The raw data of RNA-seq number is BioProject: PRJNA1030294. It can be downloaded at “Sequence read archive” on the National Center for Biotechnology Information (https://www.ncbi.nlm.nih.gov/bioproject/1030294).
Reviewer 1: The authors mention in the Abstract and Conclusion that statistically significant differences were observed amongst thawing at 37°C and 100°C, however no statistical analysis is detailed in the actual main body of the manuscript, and I see no graph or figure depicting the values for such conclusion. Please, explain further.
Finally, please revise the list of references according to the instruction for authors.
Authors: Thank you very much. The list of references now presented in one style through EndNote.
Regarding statistical treatment you are absolutely right. The mentions "statistically significant", "not statistically significant" and so on were removed from the text.

Reviewer 2 Report
Comments and Suggestions for Authors
Thank you very much for this valutable and very interesting manuscript
I have no suggestions.
Thank you very much
Additional comment:
1. What is the main question addressed by the research?The main question addressed by the research is: explore the epigenetic differences of cryopreserved mature and immature testicular tissue subjected to different thawing methods by RNA sequencing, in particular evaluating quick Thawing (in boiling water by 100°C) and slow thawing by physiological temperature 37° C) .
2. Do you consider the topic original or relevant in the field? Does it address a specific gap in the field?I consider the topic original and relevant in the field, and it address a specific gap in the field, as it is the first study in human being.
3. What does it add to the subject area compared with other published material?It is the first study in human being in this specific field of research.
4. What specific improvements should the authors consider regarding the methodology? What further controls should be considered?I have no specific suggestions as regards potential specific improvements in methodology.
5. Are the conclusions consistent with the evidence and arguments presented and do they address the main question posed?The conclusions are consistent with the evidence and arguments presented and they address the main question posed
6. Are the references appropriate?The references are appropriate
7. Please include any additional comments on the tables and figures.Figures are ok
Author Response
Answer to Reviewer 2.
Reviewer 2: Thank you very much for this valutable and very interesting manuscript
I have no suggestions.
Additional comment:
- What is the main question addressed by the research?
The main question addressed by the research is: explore the epigenetic differences of cryopreserved mature and immature testicular tissue subjected to different thawing methods by RNA sequencing, in particular evaluating quick Thawing (in boiling water by 100°C) and slow thawing by physiological temperature 37°C).
- Do you consider the topic original or relevant in the field? Does it address a specific gap in the field?
I consider the topic original and relevant in the field, and it address a specific gap in the field, as it is the first study in human being.
- What does it add to the subject area compared with other published material?
It is the first study in human being in this specific field of research.
- What specific improvements should the authors consider regarding the methodology? What further controls should be considered?
I have no specific suggestions as regards potential specific improvements in methodology.
- Are the conclusions consistent with the evidence and arguments presented and do they address the main question posed?
The conclusions are consistent with the evidence and arguments presented and they address the main question posed
- Are the references appropriate?
The references are appropriate
- Please include any additional comments on the tables and figures.
Figures are ok
Authors: Dear Sir/madam,
Thank you very much for your careful work with our manuscript.

Reviewer 3 Report
Comments and Suggestions for Authors
The authors of the article were trying to optimize the thawing regimes of conventional cryopreservation of mature and immature human testicular tissue by using comparative transcriptomic analyses.
Because of the English employed, the abstract can be a little challenging to read. It should be edited by a native speaker. However, the article’s advantage is the methods used. Not only the viability of cells after thawing are assessed, but the results are also supported by comparative transcriptomic analyses. However, several details have to be explained and corrected.
The authors claim that they have performed RNA-seq to explore epigenetic changes between different thawing protocols. However, RNA-seq is intended to explore only gene expression differences. They did not test, for example, differences in miRNA expression or methylation pattern, therefore they could not be sure about epigenetic fingerprint. They should use the correct terminology in relation to the transcriptome experiment.
They should mention that the small sample size is a limitation of the study.
There is no description in the Methods, whether the same tissue sample was used for quick and slow thawing, or whether one sample was used for only one protocol. I suggest them to write how many samples was included in Group 1, Group 2, Group 3 and Group 4. They should also mark the group number of each tissue sample cohort and its sample size in Figure 6.
There is no information on how much tissue was taken for RNA extraction, what extraction protocol was used, and what the RNA integrity number was.
There is a lack of specifications regarding the Illumina kit for library preparation, the RNA-seq platform, the number of SRA data and the location of the RNA-seq performed.
Page 2, line 59: Conventional cryopreservation is the most common using method for testicular tissue preservation.
Page 2, lines 61-63: Research The entire testicular fragments may provide a better understanding of the entire process of spermatogenesis and development. (maybe rephrase this sentence)
Page 2, lines 67-69: According to the point of view existing in “classical” cryobiology, the thawing mode (stage?) is the most important in the entire technology of cryopreservation of any type of cells, including testicular cells. (maybe rephrase it like this: As per the perspective prevalent in "classical" cryobiology, the thawing mode (stage?) holds crucial importance in the cryopreservation technique for all cell types, including testicular tissue cells)
Page 2, lines 70-72: It is the technologically maximum possible thawing rate at 100°C that is used in our technology for cryopreservation of testicular tissue. (According to what I understand, cryopreservation and thawing are two related but distinct procedures; the former is employed to preserve tissue, while the latter makes it ready for use when needed)
Page 2, lines 73,74: However, there are other points of view on the rate of cell thawing, according to which thawing should be carried out at physiological temperatures. (should be rephrased)
Page 2, lines 82,83: After respective thawing and removal of cryoprotectants (Figure 1), it was evaluated viability of cells. (Maybe rephrase with something like this: Following the appropriate thawing and removal of cryoprotectants, the cell viability was assessed.)
Page 3, lines 88, 89: Fragment of mature testicular tissue from patient D. in a hypertonic solution (0.5 M sucrose) during removal of cryoprotectants shows a tissue compaction as a result of cell dehydration.
Page 3, 4, lines 91-94: Demonstration photographs: the same dehydrated fragment on filter paper of immature testicular tissue from patient C. after cryopreservation. (D, E, F) Demonstration photographs: dehydrated fragments on filter paper of mature testicular tissue from patients K., P., and M. after cryopreservation.
Page 11, line 200: …protection. (Maybe preservation is a more suitable term).
Page 11, line 201: … negation … (Do you mean negative?)
Page 13, line 315: … coolant … (Do you mean thawing agent/media?)
Page 13, lines 325, 326: According to our calculations, this results in a 15-20% increase in the thawing rate.
Page 14, lines 361-363: What was the size of collected testicular fragments and fragments subjected to equilibration?
Page 15, line 373: Please explain what you mean by ice reaching 2 to 1 mm apex.
Page 15, line 378: … 15 min., it was …
Author Response
Answer to Reviewer 3:
Reviewer 3: The authors of the article were trying to optimize the thawing regimes of conventional cryopreservation of mature and immature human testicular tissue by using comparative transcriptomic analyses.
Because of the English employed, the abstract can be a little challenging to read. It should be edited by a native speaker. However, the article’s advantage is the methods used. Not only the viability of cells after thawing are assessed, but the results are also supported by comparative transcriptomic analyses. However, several details have to be explained and corrected.
Reviewer 3: The authors claim that they have performed RNA-seq to explore epigenetic changes between different thawing protocols. However, RNA-seq is intended to explore only gene expression differences. They did not test, for example, differences in miRNA expression or methylation pattern, therefore they could not be sure about epigenetic fingerprint. They should use the correct terminology in relation to the transcriptome experiment.
There is no information on how much tissue was taken for RNA extraction, what extraction protocol was used, and what the RNA integrity number was.
There is a lack of specifications regarding the Illumina kit for library preparation, the RNA-seq platform, the number of SRA data and the location of the RNA-seq performed.
They should mention that the small sample size is a limitation of the study.
There is no description in the Methods, whether the same tissue sample was used for quick and slow thawing, or whether one sample was used for only one protocol. I suggest them to write how many samples was included in Group 1, Group 2, Group 3 and Group 4. They should also mark the group number of each tissue sample cohort and its sample size in Figure 6.
Authors: Thank you very much. Thank you for your right suggestion. Now in our manuscript we have not used the word "epigenetic".
Now in our manuscript we have written (in Introduction, Materials and Methods as well as in Discussion):
The testicular transcriptome represents the sum of all transcripts expressed by different cell groups in testicular tissue. Therefore, RNA sequencing allows better to study the entire process of spermatogenesis and development of testicular cells. Previously published results, included sequencing data, showed that the testis is the organ with the most tissue-specific genes [7, 8]. However, there are relatively few RNA sequencing reports on cryopreserved testicular tissue, and only reports on cryopreserved testicular tissue of mice and cats [9, 10] are published.
The technologically maximal possible thawing rate can be realized with the thawing of cells in boiling water (at 100°C). Just this mode of thawing was used in our experiments in comparison with the thawing at physiological temperature (37°C).
Mature testicular tissues were collected from 6 adults, immature testicular tissues were collected from 2 children. In each experimental group it was used 3 samples.
All patients were undergoing testicular biopsy after diagnosis of azoospermia as well as for fertility preservation before initiating any therapy carrying a high risk of permanent infertility, such as high-dose chemotherapy. Testicular tissue was obtained from 2 boys and 6 adults aged 3 and 5 and from 34 to 41, respectively.
The procedure of extraction of testicular tissue has been previously described in detail [43-51]. Briefly, a midline incision was made in scrotum and testis with spermatic cord was removed preferably from the hemiscrotum with the larger testis. Tunica vaginalis was opened and tunica albuginea was visualized. Under an operating microscope, tunica albuginea was widely opened in an equatorial plane preserving the subtunical vessels. After opening of tunica albuginea, testicular parenchyma was examined directly at 12-fold magnification under the operating microscope. Small samples (1–18 mg) were excised by pulling out larger, more opaque tubules from surrounding Leydig cell nodules or hyperplasia in the testicular parenchyma.
Cryopreservation of testicular tissue (Figure 1) was performed according to the previously published protocol for human ovarian tissue [52-61].
Each sample of testicular tissue was used for RNA extraction with the Trizol method. It was detected that the RIN/RQN of all samples was greater than 4.
Strand specific transcriptome library construction was completed by enriching mRNA from total RNA, sequenced by DNBSEQ high-throughput platform, and followed by bioinformatics analysis. Library was validated on the Agilent Technologies 2100 bioanalyzer. The library was amplified with phi29 to make DNA nanoball (DNB) which had more than 300 copies of one molecular. The DNBs were load into the patterned nanoarray and single end 50 (pair end 100/150) bases reads were generated in the way of combinatorial Probe-Anchor Synthesis (cPAS). RNA-seq analysis was performed using the Dr. Tom System (https://biosys.bgi.com). The raw data of RNA-seq number is BioProject: PRJNA1030294. It can be downloaded at “Sequence read archive” on the National Center for Biotechnology Information (https://www.ncbi.nlm.nih.gov/bioproject/1030294).
Reviewer 3:
Page 2, line 59: Conventional cryopreservation is the most common using method for testicular tissue preservation.
Authors: Thank you. It is corrected.
Reviewer 3: Page 2, lines 61-63: Research The entire testicular fragments may provide a better understanding of the entire process of spermatogenesis and development. (maybe rephrase this sentence)
Authors: Thank you very much for your suggestion. Now we have written: The testicular transcriptome represents the sum of all transcripts expressed by different cell groups in testicular tissue. Therefore, RNA sequencing allows better to study the entire process of spermatogenesis and development of testicular cells. Previously published results, included sequencing data, showed that the testis is the organ with the most tissue-specific genes [7, 8]. However, there are relatively few RNA sequencing reports on cryopreserved testicular tissue, and only reports on cryopreserved testicular tissue of mice and cats [9, 10] are published.
Reviewer 3: Page 2, lines 67-69: According to the point of view existing in “classical” cryobiology, the thawing mode (stage?) is the most important in the entire technology of cryopreservation of any type of cells, including testicular cells. (maybe rephrase it like this: As per the perspective prevalent in "classical" cryobiology, the thawing mode (stage?) holds crucial importance in the cryopreservation technique for all cell types, including testicular tissue cells)
Authors: Thank you very much for your suggestion. Now we have written: As per the perspective prevalent in "classical" cryobiology, the thawing mode holds crucial importance in the cryopreservation technique for all cell types, including testicular tissue cells)
Reviewer 3: Page 2, lines 70-72: It is the technologically maximum possible thawing rate at 100°C that is used in our technology for cryopreservation of testicular tissue. (According to what I understand, cryopreservation and thawing are two related but distinct procedures; the former is employed to preserve tissue, while the latter makes it ready for use when needed)
Page 2, lines 73,74: However, there are other points of view on the rate of cell thawing, according to which thawing should be carried out at physiological temperatures. (should be rephrased)
Authors: Thank you very much for your suggestion. Now we have written: The technologically maximal possible thawing rate can be realized with the thawing of cells in boiling water (at 100°C). Just this mode of thawing was used in our experiments in comparison with the thawing at physiological temperature (37°C).
Reviewer 3: Page 2, lines 82,83: After respective thawing and removal of cryoprotectants (Figure 1), it was evaluated viability of cells. (Maybe rephrase with something like this: Following the appropriate thawing and removal of cryoprotectants, the cell viability was assessed.)
Authors: Thank you very much for your suggestion. Now we have written: Following the appropriate thawing and removal of cryoprotectants (Figure 1), the cell viability was assessed.
Reviewer 3: Page 3, lines 88, 89: Fragment of mature testicular tissue from patient D. in a hypertonic solution (0.5 M sucrose) during removal of cryoprotectants shows a tissue compaction as a result of cell dehydration.
Authors: Thank you very much for your suggestion. Now we have written: Fragment of mature testicular tissue from patient D. in a hypertonic solution (0.5 M sucrose) during removal of cryoprotectants shows a tissue compaction as a result of cell dehydration.
Reviewer 3: Page 3, 4, lines 91-94: Demonstration photographs: the same dehydrated fragment on filter paper of immature testicular tissue from patient C. after cryopreservation. (D, E, F) Demonstration photographs: dehydrated fragments on filter paper of mature testicular tissue from patients K., P., and M. after cryopreservation.
Demonstration photographs: the same dehydrated fragment on filter paper of immature testicular tissue from patient C. after cryopreservation. (D, E, F) Demonstration photographs: dehydrated fragments on filter paper of mature testicular tissue from patients K., P., and M. after cryopreservation.
Authors: Thank you very much for your suggestion. Now we have written: (C1) Demonstration photographs: the same dehydrated fragment on filter paper of immature testicular tissue from patient C. after cryopreservation. (D, E, F) Demonstration photographs: dehydrated fragments on filter paper of mature testicular tissue from patients K., P., and M. after cryopreservation.
Page 11, line 200: …protection. (Maybe preservation is a more suitable term).
Page 11, line 201: … negation … (Do you mean negative?)
Page 13, line 315: … coolant … (Do you mean thawing agent/media?)
Page 13, lines 325, 326: According to our calculations, this results in a 15-20% increase in the thawing rate.
Authors: Thank you very much. All is corrected.
Reviewer 3: Page 14, lines 361-363: What was the size of collected testicular fragments and fragments subjected to equilibration?
Authors: Thank you very much for your suggestion. Now we have written: Small samples (1–18 mg) were excised by pulling out larger, more opaque tubules from surrounding Leydig cell nodules or hyperplasia in the testicular parenchyma. Cryopreservation of testicular tissue (Figure 1) was performed according to the previously published protocol for human ovarian tissue [52-61].
Figure 1 demonstrates the sizes of pieces.
Reviewer 3: Page 15, line 373: Please explain what you mean by ice reaching 2 to 1 mm apex.
Authors: Thank you very much for your suggestion. Now we have written: as soon as the ice reached size 2 to 1 mm
Page 15, line 378: … 15 min., it was …
Authors: it is corrected. Again many thanks for your help.
